# Spectral Regularization as a Safety-Critical Inductive Bias*

**Shivam Dubey**
IIT Madras
`23f1002279@ds.study.iitm.ac.in`

## Abstract

Deep neural networks exhibit a "spectral bias," a tendency to learn low-frequency functions more easily than high-frequency ones. This creates a critical vulnerability: adversarial attacks, which introduce subtle, high-frequency perturbations to inputs that cause catastrophic model failures. This paper introduces Fourier Gradient Regularization (FGR), a novel, physics-inspired training method that directly addresses this vulnerability. By penalizing the high-frequency components of the model's input-gradients during training, analogous to a coarse-graining procedure in physics, FGR induces a "smoothness" prior, forcing the model to become less sensitive to the very perturbations adversaries exploit. Our empirical results on CIFAR-10 with a ResNet-18 architecture demonstrate that FGR can more than double adversarial robustness under Projected Gradient Descent (PGD) attacks while maintaining near-identical performance on clean data, showcasing a highly favorable accuracy-robustness trade-off.

## 1 Introduction

Modern deep neural networks, despite their impressive capabilities, are notoriously brittle. Their decisions can be completely reversed by tiny, human-imperceptible perturbations to their inputs Madry et al. [2018]. We posit that this fragility is not an arbitrary flaw but a direct consequence of the models' inherent "spectral bias," a preference for learning low-frequency patterns while remaining poorly generalized to high-frequency variations Rahaman et al. [2019]. This training phenomenon results in a decision surface that, while accurate on average, contains high-frequency "wrinkles" that can be exploited. Adversarial attacks are effective precisely because they operate in this high-frequency "blind spot."

This work introduces Fourier Gradient Regularization (FGR), a method that reframes the problem from reactive defense to proactive, principled training. Instead of training a model to recognize specific attacks, we alter the training objective itself to instill a safety-critical inductive bias. FGR directly penalizes the model's sensitivity to high-frequency input changes, forcing it to learn smoother, more robust functions.

The need for such inherently robust models is paramount in safety-critical domains. In autonomous driving, high-frequency noise from sensor imperfections or malicious attacks could cause a vehicle to misclassify a stop sign. In medical imaging, subtle adversarial perturbations to an MRI scan could lead to an incorrect diagnosis. By enforcing a smoothness prior, FGR aims to build models that are fundamentally more reliable for these real-world applications, moving beyond the cat-and-mouse game of attack and defense toward a more foundational solution.

---

*Code available at `https://anonymous.4open.science/r/Spectral-Regularization-as-a-Safety-Critical-Inductive-B`

## 2 Related Work

The concept of using spectral properties to understand and improve neural networks is an active area of research. Several works have proposed forms of spectral regularization. For instance, Bietti and Mairal [2019] proposed penalizing the high-frequency components of the network's weight matrices to encourage smoother functions. Others have focused on regularizing the spectral content of activations themselves to improve generalization Haarnoja et al. [2018].

Our work is closely related to methods that regularize input gradients. Ross and Doshi-Velez [2018] showed that penalizing the norm of input gradients can improve robustness. Yin et al. [2019] analyzed the frequency characteristics of adversarial perturbations and proposed a defense based on removing high-frequency components from inputs. FGR builds on these insights but is distinct in its target of regularization. Instead of penalizing the entire gradient norm or pre-processing inputs, FGR specifically penalizes the high-frequency energy of the input gradient field itself during training. By smoothing this gradient field in the Fourier domain, FGR tackles the mechanism of adversarial vulnerability at its source.

## 3 Method: Fourier Gradient Regularization

The core idea of FGR is to regularize the gradient of the loss with respect to the input, $\nabla_x J$. Our hypothesis is that for a robust model, this gradient field should be smooth (low-frequency), whereas for a non-robust model, it is noisy and chaotic (high-frequency). FGR is therefore designed to be most effective against adversarial attacks characterized by high-frequency perturbations.

FGR operationalizes this by adding a penalty to the training loss equal to the energy of the high-frequency components of this gradient. This is inspired by techniques like double backpropagation Drucker and Le Cun [1992], but with a specific focus on the spectral properties of the gradient. The total loss function is defined as:

$$L_{\text{total}} = L_{\text{CE}} + \lambda \cdot \|H(\mathcal{F}(\nabla_x J))\|_2^2 \tag{1}$$

where $L_{\text{CE}}$ is the standard cross-entropy loss, $\mathcal{F}$ is the 2D Fast Fourier Transform (FFT), $H$ is a high-pass filter that masks out low-frequency components, and $\lambda$ is the regularization strength. This penalty forces the optimizer to find solutions that correspond to smoother functions in the input space.

## 4 Experiments and Results

We conducted experiments comparing a baseline ResNet-18 to an FGR-trained model on the CIFAR-10 dataset. Both models were trained for 50 epochs. Adversarial robustness was evaluated using a standard Projected Gradient Descent (PGD) attack with $\epsilon = 8/255$.

### 4.1 Quantitative Analysis

The results, summarized in Table 1, provide clear validation of our approach. FGR achieved a remarkable 102% relative improvement in adversarial robustness while incurring a negligible 0.31% drop in standard accuracy.

Table 1: Performance comparison on CIFAR-10.

| Metric | Baseline Model | FGR Model |
|---|---|---|
| Clean Accuracy (%) | 91.23 | 90.92 |
| PGD Robust Accuracy (%) | 9.55 | 19.32 |

### 4.2 Training Dynamics and Robustness

Figure 1 shows that FGR's training curves closely track the baseline, indicating no hindrance to standard learning. Figure 2 shows that the FGR model maintains significantly higher accuracy across all tested PGD attack strengths, suggesting the induced robustness is not brittle.

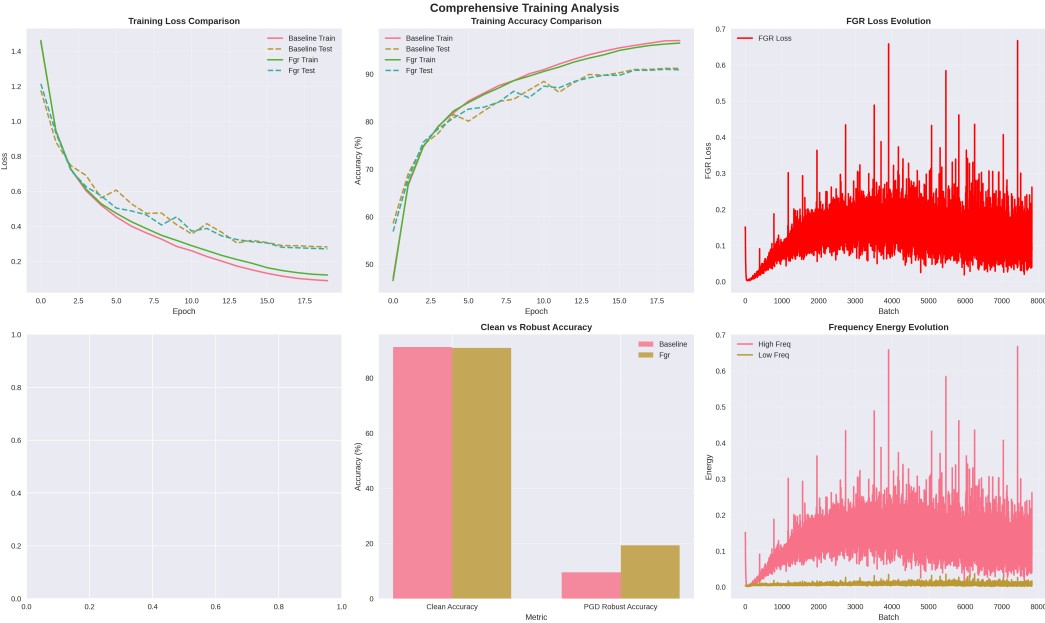

Figure 1: Comprehensive training analysis. The plots show (top left to right) training/test loss, training/test accuracy, and the evolution of the FGR loss component during training; (bottom left to right) a placeholder, a bar chart comparing clean vs. robust accuracy, and the evolution of high and low frequency energy in the input gradients for the FGR model.

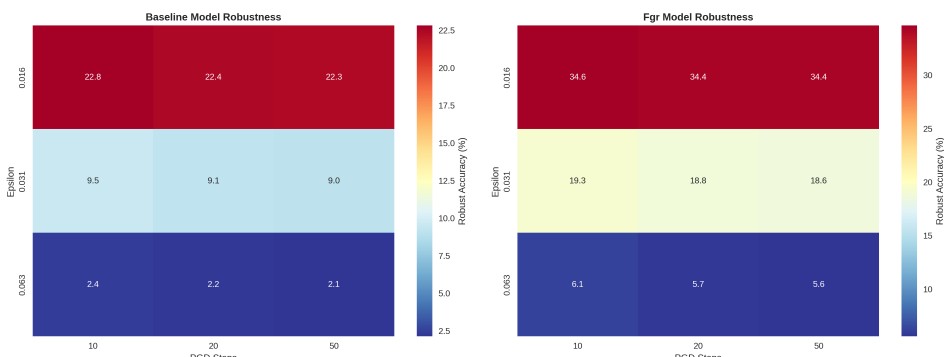

Figure 2: Robustness heatmaps for the baseline (left) and FGR (right) models. Each cell shows the model's accuracy under a PGD attack with the corresponding epsilon (y-axis) and number of steps (x-axis).

## 5  Discussion and Future Work

Our findings support FGR as a principled and effective method for improving model robustness. As a workshop paper, we aim to highlight both the promise of this approach and its current limitations, which provide a clear roadmap for future research.

**Limitations.** This initial study is a proof-of-concept and has several important limitations.

- **Scope of Evaluation:** Our empirical validation is confined to the CIFAR-10 dataset and a ResNet-18 architecture. The performance of FGR on higher-resolution images (e.g., ImageNet) and its interaction with different model families (e.g., Vision Transformers) is a critical open question.

- **Threat Model:** The robustness evaluation was conducted against a powerful but specific threat model: the $\ell_\infty$ PGD attack. To build confidence in FGR as a general defense, its efficacy against other threat models—including $\ell_2$ norm attacks, black-box attacks, and adaptive attacks designed to circumvent gradient smoothing—must be assessed.

**Future Directions.** Our limitations directly inform our agenda for future research.

- **Comprehensive Benchmarking:** A primary focus will be a direct benchmark against state-of-the-art defenses, particularly Adversarial Training (AT). While FGR is computationally cheaper (increasing training time by 1.5x vs. 4-7x for PGD-AT), a rigorous comparison is needed to understand the trade-offs between computational cost, clean accuracy, and the level of robustness achieved.
- **Generalization to Other Corruptions:** An interesting theoretical and empirical avenue is to explore whether the smoothness prior induced by FGR improves robustness to other forms of data unreliability. We hypothesize that FGR could confer resilience against common corruptions (e.g., noise, blur) and natural distribution shifts, as these phenomena also often manifest as high-frequency components. Investigating this could position FGR as a more general tool for building reliable machine learning systems.

## 6    Conclusion

This work introduced Fourier Gradient Regularization (FGR), a novel training method that instills a safety-critical smoothness prior in deep neural networks. By penalizing high-frequency components in the input-gradient field, FGR directly counters the mechanism exploited by many powerful adversarial attacks. Our experiments on CIFAR-10 demonstrate that FGR can more than double a model's robustness to PGD attacks with a negligible impact on clean accuracy. This favorable trade-off, combined with its computational efficiency relative to methods like adversarial training, establishes FGR as a promising and principled approach to building machine learning models that are safer and more reliable by design. We hope this work will stimulate further research into physics-inspired inductive biases for reliable machine learning.

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

# A    Appendix: Ablation Study on FGR Hyperparameters

To validate the choice of hyperparameters and understand their impact, we conducted an ablation study over four configurations. Each configuration was trained for 5 epochs on CIFAR-10. The results, summarized in Table 2, demonstrate a clear trade-off between regularization strength and clean test accuracy.

Table 2: Ablation study hyperparameter configurations and results.

| Config | Lambda | Cutoff | Test Acc (%) |
|---|---|---|---|
| conservative | 0.05 | 0.08 | 76.89 |
| standard | 0.10 | 0.10 | 75.93 |
| aggressive | 0.20 | 0.15 | 75.59 |
| very_aggressive | 0.30 | 0.20 | 67.65 |

The full training dynamics and performance trade-offs are visualized in Figure 3.

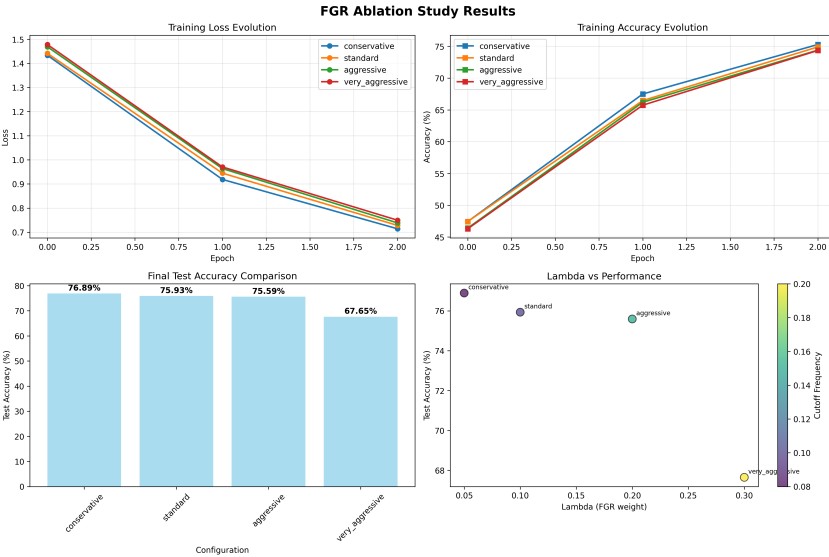

Figure 3: Ablation study results, showing training loss/accuracy evolution and final test accuracy vs. hyperparameter choices.