# OpenReview forum: "Spectral Regularization as a Safety-Critical Inductive Bias"
_NeurIPS.cc/2025/Workshop/Reliable_ML — NeurIPS 2025 - Reliable ML Workshop_

### Official Review · Reviewer_UAjt · 2025-09-17
**This paper claims that by penalizing the high frequency components of the gradients, we can train the model such that it will be immune to high frequency adversarial attacks.**

**Rating:** 6
**Confidence:** 4

**Review:**

Although this paper presents a very interesting research direction that draws inspiration from Fourier transform, it lacks a lot of details.
1) It doesn't define the high-pass filter function (H). I am curious what that high-pass filter function might look like in the time domain. How can we compare usual regularizers to that function.
2) The paper seems to have been put together in a rush; a lot of experimental details are not explained well and are hard to understand for me.
3) I would suggest improving the presentation and theoretical details further. Also add more empirical studies.

---

### Official Review · Reviewer_kLSJ · 2025-09-20
**Review for "Spectral Regularization as a Safety-Critical Inductive Bias"**

**Rating:** 6
**Confidence:** 3

**Review:**

The paper studies the problem of adversarial attacks against deep neural networks. Specifically, the authors posit that the sensitivity that many models have against specific high-frequency attacks is due to a spectral bias in the model training, making it more sensitive to high-frequency components of the input gradient field, at the expense of low-frequency components. To combat this, the authors introduce the method of Fourier Gradient Regularization (FGR), which penalizes high-frequency components during training to encourage smoother, more robust models. They test their method on the CIFAR-10 dataset, displaying a ~10% improvement in accuracy against PGD attacks with only a negligible decrease in accuracy on clean data.

Strengths

- A main strength of the paper's approach is its simplicity. It essentially says: "we have an issue by relying too much on high-frequency components, so let's simply rely less on them and see if we can get the parameters to work", and somewhat surprisingly, it does.

- While previous research has focused on regularizing input gradients or pre-processing inputs, this paper's technique of directly smoothing the gradient field in the Fourier domain seems like a new idea.

Weaknesses

- As the authors rightly point out, the study's scope is a limitation; the evaluation is confined to the CIFAR-10 dataset with a ResNet-18 architecture and tested against a specific PGD attack.

- Also, I am generally surprised that one is not giving up something by ignoring high-frequency components of the input gradient field. My personal opinion is that there's no free lunch: either the accuracy cannot be improved much more than ~20% against PGD attacks, or there is a clean dataset out there that the method performs quite poorly on.

Having said that, I like the direct approach and the paper's idea definitely needs to be explored more to see whether or not it breaks at some point. Overall, this paper is a compelling proof-of-concept for defending against PGD attacks in an era where this is of vital importance. The authors are transparent about the current limitations and lay out a clear path for future research, including more comprehensive benchmarking and testing against a wider variety of threat models and data corruptions.